# Correlation of Neutrophil-to-Lymphocyte Ratio and the Dilation of the Basilar Artery with the Potential Role of Vascular Compromise in the Pathophysiology of Idiopathic Sudden Sensorineural Hearing Loss

**DOI:** 10.3390/jcm11195943

**Published:** 2022-10-08

**Authors:** Dae-Woong Kang, Seul Kim, Woongsang Sunwoo

**Affiliations:** 1Department of Otorhinolaryngology, Seoul National University Hospital, Seoul National University College of Medicine, Seoul 03080, Korea; 2Department of Otorhinolaryngology, Gil Medical Center, Gachon University College of Medicine, Incheon 21565, Korea

**Keywords:** sudden sensorineural hearing loss, neutrophil-to-lymphocyte ratio, etiology, basilar artery

## Abstract

Idiopathic sudden sensorineural hearing loss (SSNHL) currently lacks a clear etiology, as well as an effective treatment. One of the most probable explanations for SSNHL is impairment of the cochlear blood flow. However, dissimilar to a fundoscopic examination, direct observation of cochlear blood vessels is not possible. To indirectly support an ischemic etiology of SSNHL, we investigated whether the degree of initial hearing loss is associated with two atherosclerotic risk factors: dilatation of the basilar artery (BA) and a chronic subclinical inflammatory status measured by the neutrophil-to-lymphocyte ratio (NLR). This retrospective study collected data from 105 consecutive patients diagnosed with idiopathic SSNHL. Then, the patients were divided into two groups according to their NLR as “abnormally high NLR (>3.53, *n* = 22)” and “NLR within the normal range (0.78–3.53, *n* = 83)”. The BA diameter and severity of initial hearing loss were significantly correlated with each other in the abnormally high NLR group (*p* < 0.001). However, there was no significant correlation between initial hearing loss and the BA diameter in the normal NLR group (*p* = 0.299). Therefore, the NLR may serve as a marker for SSNHL of vascular etiology and a rationale for magnetic resonance imaging examinations based on the pathophysiology.

## 1. Introduction

Sudden sensorineural hearing loss (SSNHL) is defined as sensorineural hearing loss of 30 dB or more at three consecutive frequencies within 3 days and is often accompanied by tinnitus, dizziness, ear fullness, nausea, and vomiting [1]. Although the annual prevalence of sudden hearing loss has been reported to be 5–20 per 100,000, the actual prevalence is expected to be higher, considering that the spontaneous recovery rate is approximately 32–65% [2,3]. Viral or bacterial infection, vascular disturbance, microcirculatory failure, trauma, autoimmune disease, and neoplasm are possible etiologies of SSNHL [4,5]. However, most cases of SSNHL have no identified etiology and are commonly termed idiopathic.

Circulatory disturbance has been speculated to be a possible cause of SSNHL, because the cochlea receives a limited blood supply without collateral circulation. The inner ear is an end organ supplied only by the cochlear artery, which branches out from the internal auditory artery (IAA). As expected from this vulnerable anatomical condition, the occurrence of unilateral SSNHL has been reported as an early presentation of ischemic stroke of the anterior inferior cerebellar artery (AICA), where the IAA is branched [6,7]. Consistent with these case reports, animal studies have shown that disruption of the cochlear blood flow can immediately lead to cochlear dysfunction [8,9]. In addition, several studies have suggested that defibrinogenation therapy or adjuvant heparin therapy could be more beneficial than steroid monotherapy in some patients with profound hearing loss [10,11,12]. Accordingly, these previous findings indicate the importance of considering the vascular etiology when evaluating SSNHL of an unknown cause.

Recently, the neutrophil-to-lymphocyte ratio (NLR) has been widely used not only as an inflammatory marker but also as a predictive marker in various diseases, including diabetes mellitus, chronic pulmonary disease, autoimmune disease, and cardiovascular disease [13,14,15,16]. Several studies have also investigated the association between NLR and SSNHL. These studies revealed that a high NLR is associated with the development and poor prognosis of SSNHL [17,18,19,20]. However, the reason a high NLR is correlated with a higher risk in patients with SSNHL remains unknown. To the best of our knowledge, the detailed role of the NLR in the pathogenesis of SSNHL has not yet been studied. Therefore, there is a need for evidence to elucidate the role of an elevated NLR in the high incidence and poor outcome of SSNHL. Based on the relationship between an elevated NLR and atherosclerotic events in cardiovascular disease, it has been postulated that endothelial dysfunction caused by chronic subclinical inflammatory conditions may play a role in the initiation and progression of atherosclerosis [13,21]. Thus, we hypothesized that an elevated NLR may be related to a diminished cochlear blood flow or cochlear ischemia caused by endothelial dysfunction or microvascular inflammation in the blood vessels supplying the cochlea.

To investigate the etiological role of an elevated NLR in SSNHL, we focused on the vascular etiology and evaluated the radiological findings of the basilar artery (BA) as the origin of the AICA using magnetic resonance imaging (MRI). It has been suggested that morphological deformation of the BA, including angulation and dilatation, may be associated with atherogenesis in the vertebrobasilar system and contribute to the development of SSNHL by decreasing blood flow to the cochlea and generating microemboli or microthrombi [22,23,24,25]. Therefore, we aimed to investigate whether a high NLR in SSNHL is associated with BA dilatation. Specifically, we hypothesized that if the etiology of SSNHL is vascular, the degree of reduction in the cochlear blood flow may be associated with the degree of initial hearing loss.

## 2. Materials and Methods

The Institutional Review Board of Gachon University Gil Medical Center approved this study (IRB No. GFIRB2021-243) and waived the need for informed consent owing to the retrospective nature of the study and the use of anonymous clinical data for analysis. This study complied with the Declaration of Helsinki and was performed according to ethics committee approval.

This retrospective single-center study included patients diagnosed with idiopathic SSNHL at the Gachon University Gil Medical Center between January 2014 and December 2018. Patients were included if they had undergone (1) pure-tone audiometry, (2) laboratory tests including a complete blood count (CBC) with differential, and (3) an MRI of the temporal bone within 2 weeks of hearing loss onset. Patients meeting any of the following criteria were excluded: age ≤ 18 years, white blood cell (WBC) count >10,000 cells/mm^3^, and lesions on the MRI causally or potentially related to hearing loss. After exclusion, 105 patients (54 men and 51 women aged 27–81 years) were included in the final analysis.

The initial hearing levels were measured using pure-tone audiometry at the time of diagnosis. Using the average hearing threshold level at speech frequencies (500, 1000, 2000, and 4000 Hz) in the affected ear, we computed the pure-tone average (PTA) to estimate the hearing loss. The pure-tone audiogram shapes were classified as follows: “up-sloping” or “down-sloping” (minimum difference of 20 dB between the 500-Hz and 4000-Hz thresholds), “flat” (difference between the 500-Hz and 4000-Hz thresholds <20 dB), and “profound” (PTA >90 dB) [26,27].

The CBC with differential reported counts for five main types of WBCs, either as percentages or as the absolute number of cells (cells/mm^3^). The NLR was calculated by dividing the absolute neutrophil count by the absolute lymphocyte count. Based on the normal NLR values for healthy adults reported by Forget et al., an NLR > 3.53 was considered abnormal [28]. Patients were classified into two groups: those with normal NLR (≤3.5) and those with abnormally high NLR (>3.5).

All MRI data were acquired using a 3-Tesla MRI scanner (Skyra; Siemens Medical System, Erlangen, Germany). The diameter of the BA was measured at the mid-pons level on the axial view of a 3-dimensional (3D) T2-weighted image. The parameters of the T2-weighted sampling perfection with application-optimized contrasts using different flip angle evolutions sequence were as follows: repetition time, 1000 ms; echo time, 140 ms; flip angle, 120°; field of view, 208 × 230; matrix size, 384 × 345; and slice thickness, 0.8 mm. When the BA displays angulation in its course, its longitudinal axis can be tilted to the measured plane. Thus, the short axis of the elliptical BA at the level of entry of the trigeminal nerve into the anterolateral aspect of the pons was considered the diameter of the BA (Figure 1). The MRI scans were analyzed while blinded to all clinical information.

Descriptive data were reported as the median (range) or count (percentage). The chi-square test or Fisher’s exact test were used to compare categorical variables. Continuous variables were compared using the Student’s *t*-test, and groups for skewed variables were compared using the Mann–Whitney *U* test. A one-way analysis of variance (ANOVA) was performed to compare the effect of four different audiometric curves on the NLR values. Audiometric types were compared between the normal NLR and abnormally high NLR groups using a linear-by-linear association test. The Spearman rank correlation coefficient was used to evaluate the correlation between the BA diameter, NLR, and initial PTA. To test whether the effect of the initial PTA on the NLR is dependent on the value of the BA diameter, a multiple linear regression was calculated, including the interaction terms of the initial PTA and BA diameter. Statistical significance was set at *p* < 0.05. All statistical analyses were performed using IBM SPSS software (IBM Corp., Armonk, NY, USA).

## 3. Results

The demographic and general characteristics of the study population are summarized in Table 1. The mean age of the 105 patients (54 men and 51 women) in the study population was 52.1 ± 11.6 years. The mean WBC count was 6.73 ± 1.36 × 10^3^ cells/mm^3^, and the NLR value ranged from 0.8 to 11.75 (median, 2.05). The median PTA at diagnosis was 68.8 dB (range: 20–120 dB). The flat type (*n* = 38, 36.2%) was the most common configuration of the audiogram, followed by the up-sloping (*n* = 23, 21.9%) and down-sloping (*n* = 18, 17.2%) types. Twenty-six patients with PTA > 90 dB (24.8%) were classified as the profound type. There were no statistically significant differences in the NLR values between groups, as determined by one-way ANOVA (*F*(3, 101) = 0.508, *p* = 0.678). The median BA diameter was 3.44 mm (range, 2.29–5.16 mm). There were only five cases (4.8%) of BA dolichoectasia in which the diameter of the BA was >4.5 mm.

There was no correlation between the NLR and initial PTA and BA diameter (Figure 2). Specifically, the Spearman correlation coefficient of the NLR with the initial PTA and BA diameter was *r*_S_ = −0.003 (*p* = 0.976) and *r*_S_ = −0.027 (*p* = 0.781), respectively.

Patients were divided into two groups according to their NLR. Twenty-two patients had an abnormally high NLR value (NLR > 3.53), and eighty-three had an NLR value within the normal range (NLR between 0.78 and 3.53) [28]. A comparison of the general data between the two groups is shown in Table 1. In the univariate analysis, patients with vertigo showed a higher probability of having an abnormally high NLR (*p* = 0.040). None of the other clinical characteristics, including the initial PTA and BA diameter, were statistically associated with the NLR.

A multiple linear regression was run to predict the NLR from age, sex, initial PTA, BA diameter, and the two-way interaction term of the initial PTA and BA diameter. These variables statistically significantly predicted the NLR, *F*(5, 99) = 1.44, *p* = 0.216, *R*^2^ = 0.068. Age and sex did not significantly predict the NLR (*p* = 0.9 and 0.2, respectively). The initial PTA and BA diameter were significant predictors of the NLR (*p* = 0.044 and 0.049, respectively). It was found that the interaction between the initial PTA and BA diameter was significant (*p* = 0.042). This means that the effect of the initial PTA on NLR depends on the BA diameter and vice versa.

Assuming that the increased risk of atherosclerotic events and cochlear ischemia would be better described by the combination of the NLR and BA diameter, we analyzed the etiological value of the NLR/BA diameter combination in idiopathic SSNHL. First, we assessed the relationship between the BA diameter and the initial PTA level for each group separately. As shown in Figure 3, a significant positive correlation was observed between the initial PTA and BA diameter in the high NLR group (*r*_s_ = 0.702, *p* <0.001), suggesting that, for idiopathic SSNHL of vascular etiology, a dilated BA would increase susceptibility to an insufficient cochlear blood supply and increase the initial level of hearing loss. However, there was no significant link between the initial PTA and BA diameter in the normal NLR group (*r*_s_ = 0.115, *p* = 0.299).

Spearman’s correlation coefficient was used to examine the cutoff of NLR in predicted cases of vascular etiology based on our hypothesis. Figure 4 illustrates Spearman’s rho values between the BA diameter and the initial PTA level in the high NLR group as the NLR cutoff value varies from 1.0 to 6.0. The Spearman’s correlation coefficients were between 0.70 and 0.82, which indicate a high degree of correlation, when the NTR cutoff values of 3.44–4.60 were applied. Since there were only twelve patients with values of the NLR higher than 4.61 in this study, the higher the NLR cutoff value at 4.60 or more, the lower the degree of correlation. In addition, 3.69 was the lowest NLR value among patients with NLRs higher than 3.44, so applying 3.53 as the NLR cutoff value was considered to be appropriate.

## 4. Discussion

Various potential etiological factors and a lack of helpful etiological markers in idiopathic SSNHL make a pathophysiology-based approach to the treatment of SSNHL difficult. Recently, the NLR has been proposed as an easily accessible and reliable biomarker of vascular etiology in a variety of diseases. In this study, we investigated the association between the severity of the initial hearing loss and BA diameter using temporal bone MRI and investigated the etiological value of the NLR by classifying patients with SSNHL into two groups based on normal NLR reference values. The main findings of the present study were as follows: (1) neither the degree of BA dilatation or the severity of the initial hearing loss showed a quantitative relationship with the NLR itself, and (2) in SSNHL patients with abnormally high NLR values, the initial hearing loss was more severe as the BA diameter increased, a significant correlation that was not observed in SSNHL patients with normal NLR values.

Table 1 shows the demographic and clinical characteristics of the patients with SSNHL included in this study. We obtained not only sociodemographic factors such as age and sex but also conventional risk factors associated with vascular occlusion and circulatory disturbances, including hypertension and diabetes mellitus. These factors can contribute to the vascular etiology of SSNHL, regardless of the NLR, by accelerating the prothrombotic events in the BA [22]. As presented in Table 1, because there were no significant differences in the risk factors between the high NLR group and the normal NLR group, they were not adjusted for in further analyses. In this study, the proportion of patients with vertigo was higher in the high NLR group. Since the inner ear has a blood supply from the labyrinthine artery, which branches into the anterior vestibular artery and the common cochlear artery, vestibular symptoms can occur frequently in AICA infarctions [29]. In addition, previous studies have interpreted vertigo symptoms in SSNHL to result from an ischemic etiology [30,31]. Therefore, considering that both SSNHL with vertigo and SSNHL with high NLR are related to a poor prognosis, we believe that a higher proportion of vertigo symptoms in the high NLR group arises from dysfunction of the audio–vestibular system due to vascular causes [17,32].

The severity of the initial hearing loss, measured by the initial PTA as an indicator of vascular dysfunction, was used to evaluate the vascular etiological value of the NLR in this study. Electrophysiological studies have revealed that the extent of cochlear blood flow reduction during circulatory blockage in the BA system correlates with cochlear dysfunction [9]. Based on these experiments, we assumed that the more severe the reduction in cochlear blood flow, the greater the initial PTA. Consistent with this assumption, clinical studies have reported that patients with a higher initial PTA, which means more severe hearing impairment at the time of diagnosis, had a poor response to conventional steroid therapy, resulting in poorer hearing outcomes [27,33]. Considering the possible action of steroids to reduce inflammation and edema in the hearing organs, the administration of steroids to treat patients with SSNHL caused mainly by vascular pathology may not be effective. In addition, other treatment modalities proposed to improve cochlear microcirculation based on vascular pathogenesis, including defibrinogenation, adjuvant heparin, and hyperbaric oxygen therapy, have shown potential benefits in patients with severe or profound initial hearing loss [10,11,12,34]. We believe that revascularization of the ischemic cochlea could help improve hearing in patients whose steroid treatment alone was ineffective.

The BA that supplies blood to the inner ear arises from the intersection of the two vertebral arteries (VAs). Since only 6–26% of people have both VAs with the same diameter, the BA receives an unbalanced mechanical force from the VAs on both sides [35]. This unstable and unbalanced mechanical force causes angulation and dilatation of the BA through shear stress on the vessel wall. This dilatation of the BA can cause an ischemic effect by reducing blood flow to the AICA. In addition, increased prothrombotic states in the vessel wall of the dilated BA can cause microemboli and lead to ischemia of the AICA territory [36]. Furthermore, an increased BA diameter is associated with cerebral small-vessel diseases, as well as large-vessel atherosclerosis, such as intracranial arterial stenosis [24]. Therefore, it can be expected that the BA diameter may be associated with the blood supply to the cochlea. Based on the above findings, we hypothesized that the BA diameter would be correlated with the severity of the initial hearing loss in cases of SSNHL of vascular etiology.

In the group of all subjects, significant correlations between either the BA diameter or initial PTA and the NLR were not observed. The pathophysiological heterogeneity of SSNHL may have contributed to these negative results. The inclusion of patients with SSNHL not caused by a vascular etiology may potentially render this biomarker analysis invalid. Another possible reason is the high proportion of subjects within the normal range of the NLR (83/105, 79%). In general, the prognostic and diagnostic efficacy of a marker depends on the established cutoff, which determines its effectiveness, specificity, and sensitivity. However, the optimal NLR cutoff value has not been clarified in patients with SSNHL. In previous studies, the suggested cutoff values of the NLR for predicting prognoses varied between 3.42 and 6.66 [37,38]. Accordingly, we classified all SSNHL patients into a high NLR group and a normal NLR group based on the commonly used normal reference NLR values for further analysis. However, similar to the results in the overall study population, neither the BA diameter or the initial PTA individually correlated significantly with the NLR values in either subgroup.

To improve the diagnostic efficacy of the NLR as an etiological marker in SSNHL, we used a combination of a dilated BA diameter with a high NLR value, which individually had no significant correlations with the initial hearing loss level. Interestingly, the severity of the initial hearing loss showed a significant positive correlation with the BA diameter only in the group of patients with abnormally high NLR values (Figure 3). These findings support our hypothesis that the combination of BA dilatation and an elevated NLR could indicate vascular pathogenesis in idiopathic SSNHL, such as atherosclerotic events and cochlear ischemia. As mentioned above, dilatation of the BA can reduce the blood flow through the AICA and cause ischemic damage to the cochlea. Similarly, hearing loss has been reported as one of the main symptoms in patients with BA dolichoectasia, whose BA diameter is excessively dilated by vascular remodeling [39,40]. Wang et al. suggested that mural thrombi and atherosclerotic plaques generated by slowed blood flow and turbulence in dilated and tortuous vessels could cause changes in the hemodynamics of the posterior circulation [41]. By indirectly proving the etiological value of the NLR in this study, we believe that it could be helpful to interpret the results of previous and future studies of NLR values in idiopathic SSNHL in terms of the vascular etiology.

This study had several limitations. First, our study suggests only presumptive evidence that indirectly supports the etiological value of NLR as a vascular origin in idiopathic SSNHL. To measure changes in the hemodynamics or demonstrate ischemic effects, the blood flow must be directly measured and analyzed. However, as there is still no imaging modality that can directly measure cochlear blood flow, we used an indirect method that can reflect the blood flow. Recently, several studies have reported that the use of 3D fluid-attenuated inversion-recovery MRI can provide information on pathologic conditions in the cochlea, especially intracochlear hemorrhage, a common pathological finding observed after vascular occlusion in SSNHL [8,42,43]. Therefore, further analysis of 3D fluid-attenuated inversion-recovery MRI findings combined with NLR in future studies may contribute to the elucidation of the etiological value of the NLR. Second, because the aim of this study was to evaluate the etiological value of the NLR related to vascular causes, the treatment outcomes were not analyzed. The vascular pattern in the BA system has been reported to have various anastomoses between the AICA and collateral blood vessels [44,45]. In addition, the severity of degenerative changes in the cochlea caused by vascular occlusion is affected by the presence of collateral blood vessels or anatomical variations of the IAA [8]. Therefore, this anatomical variation and the presence of collateral circulation can affect the prognosis of SSNHL of vascular etiology. Since it may not be possible to distinguish whether the outcome of hearing recovery is a response to steroid treatment or due to collateral circulation in the case of partial occlusion, analyses of the prognostic efficacy of the NLR were not conducted in the current study.

## 5. Conclusions

Idiopathic SSNHL currently lacks a clear etiology, as well as an effective treatment. One of the most probable explanations for SSNHL is impairment of the cochlear blood flow. However, dissimilar to a fundoscopic examination, direct observation of the cochlear blood vessels is not possible in patients with SSNHL. Our results indirectly support an ischemic etiology of SSNHL originating in branches of the BA and is associated with two atherosclerotic risk factors: dilatation of the BA and a chronic subclinical inflammatory status measured by the NLR. In this study, the BA diameter and severity of the initial hearing loss, which reflects the degree of reduced cochlear blood flow, were significantly correlated with each other only in patients with abnormally high NLR values. Therefore, the NLR may serve as a marker for SSNHL of vascular etiology and a rationale for an MRI examination based on the pathophysiology.

## Figures and Tables

**Figure 1 jcm-11-05943-f001:**
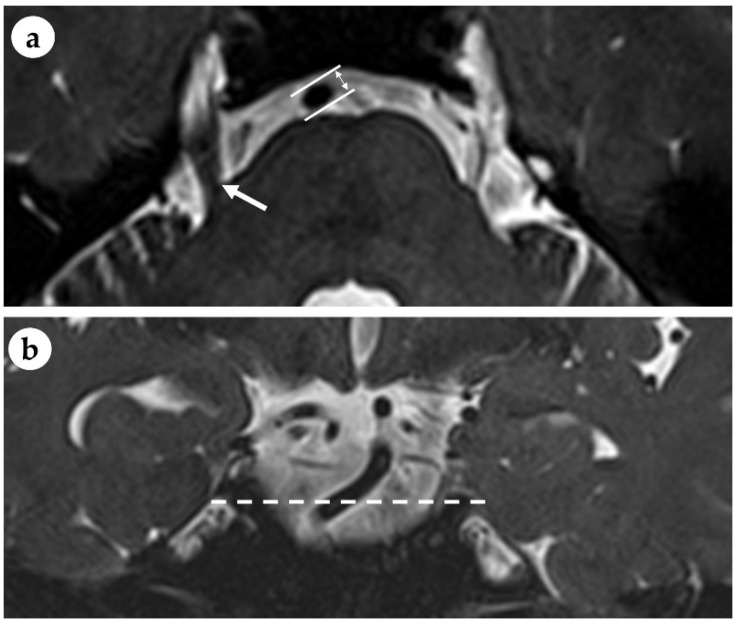
Measurement of the short axis of the basilar artery (BA) diameter (double arrow) on the axial T2-weighted magnetic resonance image at the mid-pons level; the level of the trigeminal nerve (arrow) emerging from the anterolateral aspect of the pons (**a**). Coronal T2-weighted image shows the tortuous BA that courses transversely at the mid-pons level (dotted line) (**b**).

**Figure 2 jcm-11-05943-f002:**
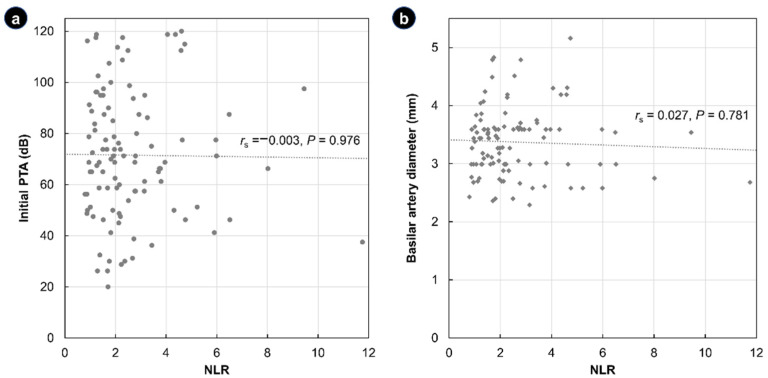
Scatter plots showing the correlation between the neutrophil-to-lymphocyte ratio (NLR) and the initial pure-tone average of the thresholds (PTA) (**a**) and the basilar artery diameter (**b**). Spearman’s correlation coefficient (*r*_s_) and corresponding *p*-value are reported alongside the regression lines.

**Figure 3 jcm-11-05943-f003:**
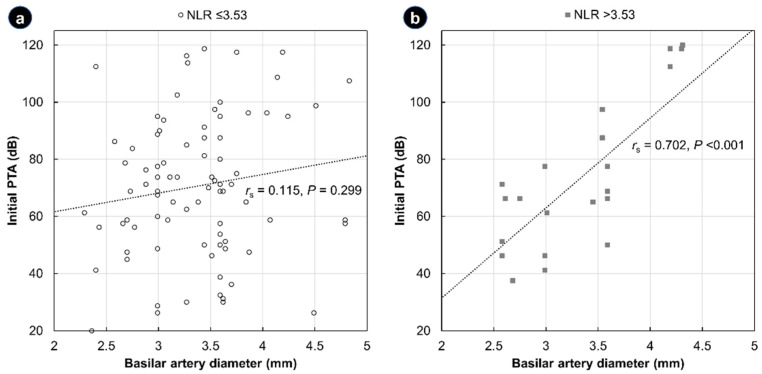
Scatter plots showing the relationship between the basilar artery diameter and the initial pure-tone average of the thresholds (PTA) in the normal neutrophil-to-lymphocyte ratio (NLR) group (**a**) and the high NLR group (**b**). Spearman’s correlation coefficient (*r*_s_) and the corresponding *p*-value are shown alongside the regression lines.

**Figure 4 jcm-11-05943-f004:**
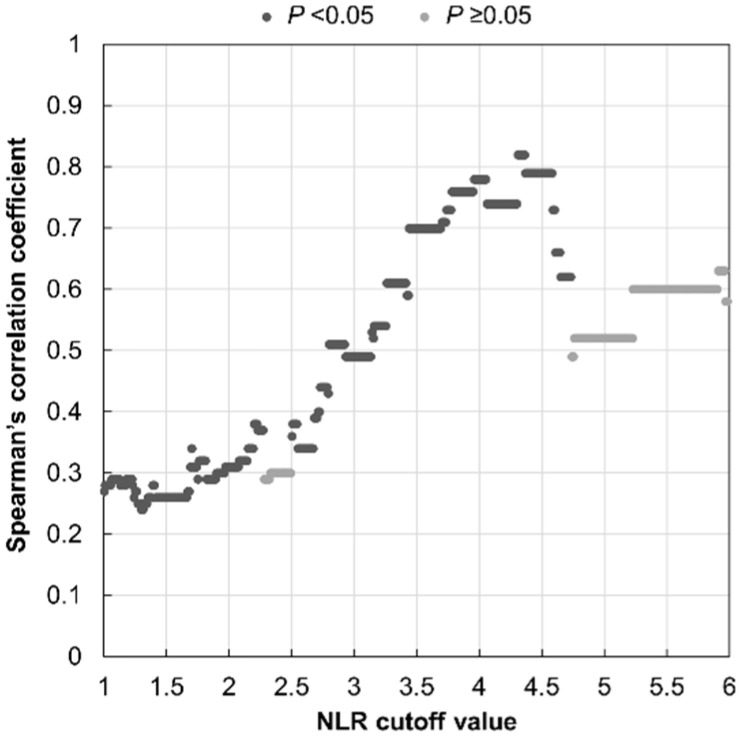
Scatter plots showing the Spearman’s correlation coefficient between the basilar artery diameter and the initial pure-tone average of the thresholds in the high neutrophil-to-lymphocyte ratio group according to the NLR cutoff value. Dark gray marks represent statistical significance (*p* < 0.05).

**Table 1 jcm-11-05943-t001:** General characteristics of the study groups.

	Total (N = 105)	NLR ≤ 3.53 (*n* = 83)	NLR > 3.53 (*n* = 22)	*p*
Age (years)	52 (27–81)	52 (27–79)	51.5 (34–81)	0.816 ^1^
Sex (male/female)	54/51	45/38	9/13	0.267 ^2^
Body mass index (kg/m^2^)	24.8 (17.4–51.0)	24.8 (17.4–51.0)	24.2 (19.1–35.2)	0.634 ^3^
Hypertension	27 (25.7%)	62 (74.7%)	6 (27.3%)	0.851 ^2^
Diabetes mellitus	17 (16.2%)	15 (18.1%)	2 (9.1%)	0.515 ^4^
Cardiovascular disease	6 (5.7%)	5 (6.0%)	1 (4.5%)	1.000 ^4^
Affected side (right/left)	56/49	48/35	8/14	0.073 ^2^
Initial PTA (dB)	68.8 (20.0–120.0)	70.0 (20.0–118.8)	67.5 (37.5–120.0)	0.595 ^3^
Vertigo	21 (20%)	13 (15.7%)	8 (36.4%)	0.040 ^4^
Audiometric curves				
Up-sloping type	23 (21.9%)	17 (20.5%)	6 (27.3%)	1.000 ^5^
Down-sloping type	18 (17.1%)	16 (19.3%)	2 (9.1%)	
Flat type	38 (36.2%)	30 (36.1%)	8 (36.4%)	
Profound	26 (24.8%)	20 (24.1%)	6 (27.3%)	
BA diameter (mm)	3.44 (2.29–5.16)	3.44 (2.29–4.83)	3.50 (2.58–5.16)	0.850 ^3^
BA dolichoectasia	5 (4.8%)	4 (4.8%)	1 (4.5%)	1.000 ^4^
WBC (×10^3^ cells/mm^3^)	6.73 (3.57–9.72)	6.57 (3.57–9.40)	7.35 (4.64–9.72)	0.015 ^1^
Neutrophil (×10^3^ cells/mm^3^)	4.21 (1.46–8.02)	3.78 (1.46–6.44)	5.83 (3.37–8.02)	<0.001 ^3^
Lymphocyte (×10^3^ cells/mm^3^)	1.96 (0.50–4.00)	2.17 (1.26–4.00)	1.15 (0.50–1.88)	<0.001 ^3^
Platelet (×10^3^ cells/mm^3^)	250.5 (130–522)	250.3 (130–522)	251.4 (148–329)	0.471 ^3^
MPV (fL)	10.1 (6.6–12.6)	10.2 (7.5–12.6)	9.8 (6.6–12.0)	0.125 ^1^

Values are presented as the median (range) or count (%). PTA, pure-tone average of the thresholds at 500, 1000, 2000, and 4000 Hz; BA, the basilar artery; NLR, neutrophil-to-lymphocyte ratio; WBC, white blood cell; and MPV, mean platelet volume. The *p*-values are computed from the ^1^ Student’s *t*-test, ^2^ chi-square test, ^3^ Mann–Whitney *U* test, ^4^ Fisher’s exact test, and ^5^ linear-by-linear association.

## Data Availability

The data that support the findings of this study are available from the corresponding author upon reasonable request.

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
