# Peer review of "Correlation of Neutrophil-to-Lymphocyte Ratio and the Dilation of the Basilar Artery with the Potential Role of Vascular Compromise in the Pathophysiology of Idiopathic Sudden Sensorineural Hearing Loss"

_jcm, 2022, doi:10.3390/jcm11195943_

Round 1

Reviewer 1 Report

This retrospective study evaluated the relationship among hearing sensitivity, neutrophil-to-lymphocyte ratio (NLR), and basilar artery (BA) diameter. The study obtained no significant association between BA and NLR ratio in the normal NLR group, and a significant association between BA and NLR ratio in the elevated NLR group. The results indicate that NLR might be a useful clinical marker for SSNHL of vascular etiology. The results are interesting, and the manuscript is written well. The following concerns should be addressed–
Major concerns:
•    Table 1 indicates that some factors (such as sex and cardiovascular conditions) might influence NLR. The study contains only 22 individuals with >3.53 NLR, which collectively makes the analysis susceptible to confounders. In addition to the analyses presented in the present study, I suggest using a multiple linear regression model (NLR as a continuous response variable) with age, sex, BA diameter, PTA, and BA diameter*PTA (interaction term). The interaction term should help test the study hypothesis, and provide further supporting evidence while controlling for the effects of potential confounders.
•    Figure 3: the data are not visualized efficiently. Please make two separate figures to show these data.
•    Table 1: add other hematological markers collected with the complete blood count. The comparison between the two groups could help illuminate other possible pathologies.
Minor concern:
Line 66: Remove “prove” and replace it with “investigate”. The measures included in the present study cannot conclusively prove the study findings. This comment applies to other places in the manuscript where the word “prove” is used.
Please avoid acronyms if the word is not used more than two-three times.

Author Response

Response to Reviewer 1 Comments

This retrospective study evaluated the relationship among hearing sensitivity, neutrophil-to-lymphocyte ratio (NLR), and basilar artery (BA) diameter. The study obtained no significant association between BA and NLR ratio in the normal NLR group, and a significant association between BA and NLR ratio in the elevated NLR group. The results indicate that NLR might be a useful clinical marker for SSNHL of vascular etiology. The results are interesting, and the manuscript is written well. The following concerns should be addressed.

We appreciate the reviewer for showing interest to our study. And thank you so much for your time and effort to review our works.

Major concerns:

Point 1: Table 1 indicates that some factors (such as sex and cardiovascular conditions) might influence NLR. The study contains only 22 individuals with >3.53 NLR, which collectively makes the analysis susceptible to confounders. In addition to the analyses presented in the present study, I suggest using a multiple linear regression model (NLR as a continuous response variable) with age, sex, BA diameter, PTA, and BA diameter*PTA (interaction term). The interaction term should help test the study hypothesis, and provide further supporting evidence while controlling for the effects of potential confounders.

Response 1: We appreciate the reviewer’s precious comment on this. We added more analysis using a multiple linear regression as suggested (line 166–172)

Point 2: Figure 3: the data are not visualized efficiently. Please make two separate figures to show these data.

Response 2: According to your valuable suggestion, Figure 3 has been changed.

Point 3: Table 1: add other hematological markers collected with the complete blood count. The comparison between the two groups could help illuminate other possible pathologies.

Response 3: According to your valuable suggestion, other hematologic markers have been added in Table 1.

Although total WBC is significantly different between the two groups, the difference is very small and does not seem to have any clinical significance. Considering the grouping based on NLR values, a higher neutrophil count and lower lymphocyte count in the high NLR group is an expected result. Finally, platelet-related markers did not differ between the two groups.

Minor concern:

Point 4: Line 66: Remove “prove” and replace it with “investigate”. The measures included in the present study cannot conclusively prove the study findings. This comment applies to other places in the manuscript where the word “prove” is used.

Response 4: Thank you for pointing it out. The word “prove” is replaced with “support” (line 15 and 292) or “investigate” (line 67).

Point 5: Please avoid acronyms if the word is not used more than two-three times.   

Response 5: According to reviewer’s valuable comment, acronyms not used more than two-three times, including CVD and FLAIR, have been removed.

Reviewer 2 Report

Authors suggested an ischemic etiology of SSNHL originating in branches of the BA and is associated with two atherosclerotic risk factors: dilatation of the BA and a chronic subclinical inflammatory status measured by NLR. This study is valuable and interesting, however there were several concerns in methodology, especially research design.

Major concerns

1. The title should be changed to “Correlation of Neutrophil-to-Lymphocyte Ratio and the dilation of BA with the Potential Role of Vascular Compromise in the Pathophysiology of Idiopathic Sudden Sensorineural Hearing Loss”.

2. Authors concluded that dilation of the BA was also important, however, authors cannot decline the possibility of collateral blood vessels. Assessment of BA is sought to be non-sense.

3. Authors’ results suggested the severity of ISSNHL, however this study should examine the incidence of ISSNHL or the prognosis od ISSNHL because auditory threshold was not decided by only subjective assessment.

4. Conclusion did not elucidate the results. Treatment should be described in Discussion.

Minor concerns

1. Authors divided the types of audiogram, however there were no mention in Results section.

2. Figure 3 and 4 were the same results. One should be deleted.

3. Authors should examine the cutoff of NLR in the ISSNHL.

Author Response

Response to Reviewer 2 Comments

Authors suggested an ischemic etiology of SSNHL originating in branches of the BA and is associated with two atherosclerotic risk factors: dilatation of the BA and a chronic subclinical inflammatory status measured by NLR. This study is valuable and interesting, however there were several concerns in methodology, especially research design.

We appreciate the reviewer for showing interest to our study. And thank you so much for your time and effort to review our works.

Major concerns

Point 1: The title should be changed to “Correlation of Neutrophil-to-Lymphocyte Ratio and the dilation of BA with the Potential Role of Vascular Compromise in the Pathophysiology of Idiopathic Sudden Sensorineural Hearing Loss”.

Response 1: We appreciate the reviewer’s valuable suggestion. According to the reviewer’s comment, the title has been changed as suggested.

Point 2: Authors concluded that dilation of the BA was also important, however, authors cannot decline the possibility of collateral blood vessels. Assessment of BA is sought to be non-sense.

Response 2: Thank you for mentioning this point. It seems to be a bit misunderstood.

Since we only included patients with idiopathic sensorineural hearing loss, of course, there were no abnormal MRI findings such as acute infart. In other words, even if the microemboli from the BA is the cause of isolated hearing loss, it can be inferred that the occlusion may occurr in the distal portion of the cochlear arterial system that mainly affects the cochlear blood flow. And, as described in the discussion section (line 249–260), we hypothesized that increased prothrombotic states in the vessel wall of the dilated BA could be a predisposing factor for vascular etiology.

In a different context, in the last paragraph of the discusion section we mentioned the possibility of the influence of collateral circulation in association with hearing outcomes. Based on previous animal experiments and human reports, in the case of total occlusion, the prognosis is not good because severe degenerative changes in the cochlea occurs in several minutes. However, in the case of partial occlusion, the variation of the collateral system can affect the hearing outcome evaluation as a confounding factor. Blood flow to the cochlea distinct from that of the AICA network or blood flow in the bone surrounding the cochlea has been considered in an explanation of the residual blood flow.

Point 3: Authors’ results suggested the severity of ISSNHL, however this study should examine the incidence of ISSNHL or the prognosis of ISSNHL because auditory threshold was not decided by only subjective assessment.

Response 3: We appreciate the reviewer’s precious comment on this. In several previous studies, a higher NLR values in ISSNHL patients compared to healthy people with normal hearing and elevated NLR values associated with poor prognosis in ISSNHL have been demonstarated. However, the incidence and prognosis themselves cannot explain the underlying mechanism related to elevated NLR values in ISSNHL. So, we focused on vascular etiology and analyzed the underlying vascular structure together with NLR. Since the association between the degree of vascular occlusion and acute auditory dysfunction has already been well reported in animal studies, we selected the severity of initial hearing impairment as a comparative variable in this study.

Point 4: Conclusion did not elucidate the results. Treatment should be described in Discussion.

Response 4: We appreciate the reviewer’s precious comment on this. According to reviewer’s comment, these sentence are rephrased to elucidate our results (line 321–323)

Minor concerns

Point 5: Authors divided the types of audiogram, however there were no mention in Results section.

Response 5: Audiometric curve type distrbution has been described in the results section and Table 1 (line 137–140). We added analysis using a one-way ANOVA to compare the effect of four different audiometric curves on NLR values (line 140–141). Because there were no significant difference in audiometric curve type dstribution according to the NLR values, further analysis was not conducted.

Point 6: Figure 3 and 4 were the same results. One should be deleted.

Response 6: Thank you for mentioning this point. According to reviewer’s valuable comment, Figure 4 and corresponding sentences have been removed.

Point 7: Authors should examine the cutoff of NLR in the ISSNHL.

Response 7: According to the reviewer’s comment, we have conducted further analysis to examine the NLR cutoff values (line 188–197, and Figure 4).

Round 2

Reviewer 2 Report

All concerns were rightly corrected.